# Zika Virus Infection Modulates Extracellular Vesicle Biogenesis and Morphology in Human Umbilical Cord Endothelial Cells: A Proteomic and Microscopic Analysis

**DOI:** 10.3390/microorganisms13061402

**Published:** 2025-06-16

**Authors:** Manuel Adrián Velázquez-Cervantes, Arturo Flores-Pliego, Yazmín Rocío Benitez-Zeferino, Victor Javier Cruz-Holguín, Luis Herrera Moro-Huitron, Addy Cecilia Helguera-Repetto, David Eduardo Meza-Sánchez, José Luis Maravillas-Montero, Nicolás Cayetano-Castro, Javier Mancilla-Ramírez, Aurora Casarrubias-Betancourt, Guadalupe León-Reyes, Macario Martínez-Castillo, Isabel Wong-Baeza, Luis Adrián De Jesús-González, María Isabel Baeza-Ramírez, Moisés León-Juárez

**Affiliations:** 1Laboratorio de Virología Perinatal y Diseño Molecular de Antígenos y Biomarcadores, Departamento de Inmunobioquimica, Instituto Nacional de Perinatología “Isidro Espinosa de los Reyes”, Mexico City 11000, Mexico; adrianvela18@gmail.com (M.A.V.-C.); bezyqcb@hotmail.com (Y.R.B.-Z.); victor.cruz@cinvestav.mx (V.J.C.-H.);; 2Laboratorio de Biomembranas, Escuela Nacional de Ciencias Biológicas, Instituto Politécnico Nacional, Mexico City 11340, Mexico; mwongb@ipn.mx (I.W.-B.); isabelbaeza@yahoo.com (M.I.B.-R.); 3Departamento de Inmunobioquimica, Instituto Nacional de Perinatología “Isidro Espinosa de los Reyes”, Mexico City 11000, Mexico; arturo_fpliego@yahoo.com.mx (A.F.-P.); addy.helguera@inper.gob.mx (A.C.H.-R.); 4Red de Apoyo a la Investigación, Coordinación de la Investigación Científica, Instituto Nacional de Ciencias Médicas y Nutrición “Salvador Zubirán”, Universidad Nacional Autónoma de México, Mexico City 04510, Mexico; dmeza@cic.unam.mx; 5Departamento de Medicina Molecular y Bioprocesos, Instituto de Biotecnología, Universidad Autónoma de México, Cuernavaca City 62210, México; maravillas@cic.unam.mx; 6Centro de Nanociencias y Micro y Nanotecnologías, Instituto Politécnico Nacional, Mexico City 07738, Mexico; ncayetanoc@ipn.mx; 7Escuela Superior de Medicina, Instiuto Politécnico Nacional, Mexico City 11340, Mexico; drmancilla@gmail.com; 8Hospital de la Mujer, Secretaria de Salud, Mexico City 11340, Mexico; dra.auro.court@gmail.com; 9Laboratorio de Nutrigenética y Nutrigenómica, Instituto Nacional de Medicina Genómica (INMEGEN), Mexico City 14610, Mexico; greyes@inmegen.gob.mx; 10Sección de Estudios de Posgrado e Investigación, Escuela Superior de Medicina, Instituto Politécnico Nacional, Mexico City 1340, Mexico; mmartinezcas@ipn.mx; 11Departamento de Inmunología, Escuela Nacional de Ciencias Biológicas, Instituto Politécnico Nacional, Mexico City 11340, Mexico; 12Unidad de Investigación Biomédica de Zacatecas, Instituto Mexicano del Seguro Social, Zacatecas 98000, Mexico; luis.dejesus@cinvestav.mx

**Keywords:** ZIKV, EVs, HUVEC, 2D electrophoresis, cryo-TEM

## Abstract

Infection with Zika virus (ZIKV) is a perinatal health problem and a vertical infection that promotes neurological fetal damage. ZIKV infects different cellular components at the maternal–fetal interface, including umbilical cord endothelial cells (HUVECs). Extracellular vesicles (EVs) are cellular components that mediate extracellular communication. Viruses have the capacity to hijack and modify the biogenesis machinery of EVs for their own benefit. The present work provides proteomic results (2D electrophoresis) that show the regulation of the expression of proteins involved in autophagy, oxidative stress, and exosome biogenesis in HUVECs infected with ZIKV. We confirmed that Alix and CD9 proteins were downregulated following the infection. Additionally, EVs isolated from infected cells showed the expression of Alix, and CD9 was increased in contrast to the mock condition. Interestingly, nanoparticle tracking and cryo-microscopy assays revealed that these EVs showed an increase in the quantity and size of ZIKV infection to differences in mock conditions. Furthermore, EVs isolated from infected cells showed infectivity, and both RNA and viral proteins were detected. Finally, our cryo-microscopy analysis revealed that the viral infection promoted morphological changes in these extracellular vesicles to identify vesicles with double and triple vesicles and electrodense and double membranes. In conclusion, our data suggest that ZIKV infection can modulate cellular factors involved in the formation and morphology of EVs in HUVECs. Furthermore, these EVs carry viral elements that may contribute to the dissemination of infection. Future studies aimed at the proteomic and lipidomic composition analyses of these EVs are needed to understand the biological implications in vertical infection.

## 1. Introduction

Zika virus (ZIKV) is a flavivirus that, in recent years, has gained significant relevance worldwide due to its tremendous impact on women in the first trimester of pregnancy. This is because it causes the congenital syndrome associated with Zika virus (CZS) in the neurological development of the fetus, which is characterized by complications such as the development of microcephaly, damage to brain tissue, and ocular alterations [1]. In addition, several previous studies have suggested that infections are associated with autoimmune problems in adults, such as Guillain–Barré syndrome [2,3]. Interestingly, robust experimental evidence has shown that ZIKV infects several placental and extraembryonic cell lineages, such as Hofbauer cells, umbilical cord endothelial cells (HUVECs), amniotic epithelium, and cytotrophoblasts, suggesting the higher possibility that these components of the maternal–fetal interface could be ZIKV during vertical infection [4,5,6,7]. In this context, ZIKV infection promotes autophagy by upregulating LC3–I, LC3–II, and beclin–1 proteins involved in autophagosome formation [8]. Interestingly, the NS1 protein is secreted by ZIKV-infected HUVEC, and this protein reaches the blood–brain barrier, which causes hyperpermeability in these endothelial cells; thus, ZIKV is suggested to reach the fetal nervous system [9].

Exosomes are part of the subpopulations of extracellular vesicles (EVs). Currently, markers are used to discriminate exosomes that can communicate their endosomal origin such as CD63, CD81, CD9, ALIX, TSG101, and HSP70, and their size is approximately 30–150 nm. They are cellular vehicles that facilitate the transport of different biomolecules involved in various cell–cell communication purposes to maintain cellular homeostasis [10]. In this regard, it has been observed that viruses hijack the biogenesis machinery of exosomes, change their composition, and, in turn, transport viral proteins, viral RNA, viral particles, and cellular factors that affect the response of other cells. This enhances viral replication in the cellular environment, in addition to producing immunity against viral pathogens by activating an antiviral response in adjacent cells [11,12].

The potential of viruses to modulate the exosome pathway is important for understanding how it influences the dissemination and pathogenicity of these viral agents, as the hijacking of the exosome biogenesis pathway by viruses directly increases the size and quantity of these EVs. Among the viruses that have been studied that sequester the biogenesis of exosomes and affect the protein cargo within these vesicles, it has been described that EVs isolated from lymphocytes infected with HIV–1 promote the death of CD4 T lymphocytes [13,14]. Meanwhile, in mosquito cells infected with DENV–2, the isolated EVs contain the protein Tsp29Fb, an orthologue of CD63 in humans, and it was observed that Tsp29Fb interacts with the envelope protein of this flavivirus [15]. This interaction promoted the replication and transmission of this virus. Another virus of this family is ZIKV, which has been reported to carry viral RNA and the envelope protein in the exosomes during infection, aiding viral dissemination [16]. The change in EVs load is not the only factor that can be observed in this context, since the size of these vesicles and their number increase during viral infections; it has been observed that rabies virus, DENV, and ZIKV itself have this modulation on EVs [17,18,19]. These data indicate that not only protein cargo changes under infection conditions, but the number of vesicles and the regulation of EVs’ sizes also change.

A poorly studied field in the context of EVs is the different morphological populations, since cryo-TEM imaging has identified a great variety of vesicles and classified them as single, oval, double, multivesicular, incomplete or broken membranes, tubular, pleomorphic, vesicular sacs, electrodense, and double membrane bilayers [20,21,22,23]. A change in the proportion of these morphological populations has been observed, as reported in the M1000 prion infection of GT1–7 mouse hypothalamus cells, with an increased number of single vesicles and a decrease in the multivesicular population [24].

The protein composition of EVs during viral infection is of great importance because it provides information about the possible enrichment of these vesicles with exosomal proteins or an increase in the number of EVs produced, as shown in rabies virus infection by the enrichment of CD63 and TSG101 EVs under infection conditions, reflected in the increase in the number of EVs [17]. In this sense, the study of the protein composition of ZIKV-infected HUVECs is of great importance to identify and understand which exosomal proteins are being regulated and the impact that this would have on the biogenesis of these EVs, both in their size and number of vesicles, as well as the behavior of the morphology of EVs during infection by viral agents. Therefore, it is of the utmost importance to understand the regulation of exosomal proteins and EVs’ morphological architecture in the context of viral infections and their impact on viral pathogenesis.

## 2. Materials and Methods

### 2.1. Isolation of Human Umbilical Cord Vein Endothelial Cells (HUVECs)

Umbilical cord samples were obtained with prior authorization and signature of informed consent by women at the end of their pregnancy who wanted to participate in the research project, within the facilities of the National Institute of Perinatology (INPer). The project was approved by the bioethics committee of INPer with the registration number 212250–3210–21007–03–16. The following inclusion criteria were considered: women in the last weeks of pregnancy; clinically healthy; without apparent infections or inflammatory processes, obesity, diabetes, or arterial hypertension; and with resolution of pregnancy by cesarean section. Umbilical cord samples were obtained with a size of 10–15 cm in length, the umbilical vein was identified for treatment with 0.025% trypsin–EDTA 1X for 30 min at 37 °C, trypsin was inhibited with fetal bovine serum and centrifuged at 2500 rpm for 10 min in order to obtain the cell pellet, and this was then cultured in EndoGRO–LS medium (Merck–Millipore, Darmstadt, Germany), supplemented with 0.2% EndoGRO–LS Supplement, 5 ng/mL epidermal growth factor (EGF), 50 ng/mL ascorbic acid, 10 mM L–Glutamine, 1 µg/mL hydrocortisone hemisuccinate, 0.75 U/mL heparan sulfate, and 5% FBS (ultracentrifuged at 100,000× *g* for 70 min to avoid exosomes from the serum).

### 2.2. ZIKV Supply and Titer

Vero cells (ATCC CCL-81^TM^) were cultured in 75 cm^2^ culture boxes with RPMI medium (GIBCO Invitrogen, Waltham, MA, USA) supplemented with 2% FBS (Corning, New York, NY, USA) and 1% antibiotic–antimycotic (Thermo Fisher, Waltham, MA, USA) until 70–80% confluence and infected with ZIKV (PRVABC59). The supernatant was collected until a cytopathic effect of the culture was observed and concentrated in centricon tubes (Millipore, Darmstadt, Alemania) at 2500 rpm for 30 min at 4 °C. For viral titration, 120,000 Vero cells were cultured in 24 well plates, serial dilutions were performed, and 1% methylcellulose semi-solid medium was added for five days. After this period, the wells were washed, and 0.1% Naftol Blue Black dye was added to the lytic plates and counted.

### 2.3. Determination of Factor VIII and ZIKV Envelope Protein by Immunofluorescence

A total of 150,000 cells were cultured on coverslips, fixed with 4% paraformaldehyde (PFA) (Thermo Fisher Scientific Chemicals, Waltham, MA, USA) for 30 min, and washed with PBS1X–0.1% tritonX100 (SIGMA, St. Louis, MI, USA) for 30 s. A permeabilizing solution (PBS1X–0.1%tritonX100 and 0.05 mg gelatin) was added for 40 min, followed by three washes with the wash solution and blocking solution (PBS1X–Triton 0.1% and 1% goat serum). Primary antibodies used were anti-sheep-FactorVIII (1:100) (FITC-conjugate) (Abcam, Waltham, MA, USA) and anti-mouse-Envelope (ZIKV) (1:100) (Millipore, Burlington, MA, USA) for 1 h, and as secondary, anti-mouse-CY3 (1:200) (GeneTex, Irvine, CA, USA) was incubated for 1 h; mounting was performed with DAPI (Vector Lab, Burlingame, CA, USA) to stain cell nuclei, which were analyzed on an Olympus IX73 phase contrast fluorescence microscope (Olympus, Tokyo, Japan); a DP74 microscope digital camera (Olympus, Tokyo, Japan), objective 10x UplanFl N 10x with filters; Leica TCS SP5 confocal microscope, Helium-Neon (HeNe) laser 543 nm, transmission power 19%, emission filter BP 560–615 nm, and scanning speed 5; UV Diode laser 405 nm, transmission power 16%, emission filter BP 420–480 nm, and scanning speed 4 (Leica, Wetzlar, Germany).

### 2.4. Isolation of Exosomes of HUVECs

HUVEC supernatants were obtained after 48 h of culture. The supernatant was centrifuged at 3000× *g* for 25 min at 4 °C; then, the supernatant was recovered and centrifuged at 10,000× *g* for 45 min at 4 °C (Beckman Coulter, Avanti J-26S Series, Brea, CA, USA). The supernatants obtained were passed through 0.22 µm filters (Millipore, Cork, Ireland), and 8 mL were placed in polypropylene tubes (Beckman Coulter, Brea, CA, USA) and then centrifuged at 48,300 rpm for 70 min at 4 °C (Beckman Coulter, Optimum XPN 100, Brea, CA, USA), after which the supernatant was decanted and the pellet obtained was retained. In the next step PBS 1X was added at 4 °C, and the ultracentrifugation conditions mentioned above were used. Finally, the tubes were decanted, and the exosome pellet was resuspended in 100 μL of 1X PBS to preserve the exosomes intact and stored at −70 °C.

### 2.5. Characterization of HUVECs and Exosomes by Flow Cytometry

To perform flow cytometry, 100,000 cells were obtained, to which 300 μL of blocking solution (PBS 1X and goat serum 1:10) was added and incubated for 1 h, washed with PBS 1X, and centrifuged at 1000 rpm for 5 min. Anti-mouse-CD144 (1:300) (Thermo Scientific, Waltham, MA, USA) and anti-rabbit-CD31 (1:500) (Abcam, Cambridge, UK) for 1 h, and the secondary antibodies anti-mouse-FITC 1:100 (Thermo Scientific, Rockford, IL, USA) and anti-rabbit-PE–Cy7 (1:00) (Thermo Scientific, Centennial, CO, USA) were added for 1 h and analyzed in the BD FACSAriaTM III kit (BD Biosciences, Franklin Lakes, NJ, USA). To characterize the exosomes, anti–CD63 magnetic beads (Invitrogen, Vilnius, Lituania) were used to concentrate them, and they were incubated overnight, and anti-CD9–APC (BioLegend, San Diego, CA, USA) and anti-CD9–FITC (BioLegend, San Diego, CA, USA) antibodies were used at a 1:100 dilution and analyzed on a flow cytometer, (BD LSRfFortessa, Franklin Lakes, NJ, USA). They were evaluated in approximately 1500 events.

### 2.6. Proteome Analysis of ZIKV-Infected HUVECs

#### 2.6.1. Protein Extraction

5 × 10^6^ HUVECs cells were cultured and infected with ZIKV virus at 5 MOI for 48 hpi; the cells were detached with scraper (Corning Incorporated COSTAR, Mexico City, Mexico), centrifuged at 1000 rpm for 5 min, and suspended with 1X PBS. Subsequently, they were centrifuged at 2000 rpm for 5 min. The commercial reagent Mem–PER (Thermo Scientific, Meridian Road, Rockford, IL, USA) was used with the protease inhibitor Halt^®^ Protease and Phosphatase inhibitor (Thermo Scientific, Waltham, MA, USA) at a concentration of 1X. The supernatant was obtained by centrifugation at 13,000× *g* for 5 min at a temperature of 4 °C. The protein concentration was determined using the Lowry method in a 96-well microplate at an absorbance of 725 nm and quantified using an albumin standard curve. The extracts were stored at −70 °C until processing. Protein extracts were cleaned with a 2D Clean–Up kit (GE Healthcare, Milwaukee, WI, USA), and proteins were precipitated from the samples to remove any contamination by salts, lipids, phenols, nucleic acids, and detergents.

#### 2.6.2. Isoelectrofocusing

Ampholytes (0.5% IPG Buffer 3–10) and 40 mM DTT were added to the protein extract. The sample was placed in contact with a gel strip with a pH 3–10 gradient. NL was incubated for 12 H at 25 °C. The strips were placed in Ettan IPGphor3 (GE Healthcare, Milwaukee, WI, USA) equipment and run according to their isoelectric points for 4–5 h.

#### 2.6.3. 2D Electrophoresis

Polyacrylamide gels (10%) were prepared; the stripping gel was rehydrated in an equilibration solution with 0.1% DTT for 15 min. After mixing by inversion, a new equilibration solution containing 2% iodoacetamide was added, and the mixture was incubated for 15 min. The strip was placed on top of the polyacrylamide gel, and the gel was run at 10 mA in a running buffer for 2 h. Finally, the gels were stained with Coomassie blue (Thermo Fischer, Vilnius, Lituania).

#### 2.6.4. In Silico Analysis

Protein spots with high and low expression levels were analyzed according to their molecular weights and isoelectric points in the PhosphoSite database.

### 2.7. Analysis of Exosomal Proteins by Western Blot

Samples were added in a Laemmli buffer (Bio–Rad, Hercules, CA, USA) and denatured at a temperature of 96 °C for 10 min, and SDS–PAGE gels at 10% and 12% were run. Samples were placed on the gels for 120 min, after which they were transferred to nitrocellulose membranes for 90 min. At the end of the transfer, the membranes were blocked for 1 H with TBS1X–Tween–skimmed milk5%, and four washes of 10 min each were performed with TBS1X–Tween (Thermo Fisher, Rockford, IL, USA). Primary antibodies used were anti-rabbit-CD9 (1:800) (Abcam, Cambridge, UK), anti-mouse-ALIX (1:1000) (NovusBio, Littleton, CO, USA), anti-rabbit–NS1 (1:1000) (Genetex, Irvine, CA, USA), and anti-rabbit-prM (1:1000)) (Genetex, Irvine, CA, USA). ZIKV was added overnight to anti-Calnexin (1:1000) (Abcam, Cambridge, UK) and anti-mouse-GAPDH (1:4000) (Santacruz, Dallas, TX, USA) for 1 h incubation, and they were subsequently washed with 1X TBS and incubated with anti-rabbit–HRP secondary antibody (1:3000) (Genetex, Irvine, CA, USA), and then anti-mouse–HRP (1:3000) (Invitrogen, Waltham, MA, USA) was added for 1 h. Finally, membranes were developed by the SuperSignal™ West Femto chemiluminescence kit (Thermo Scientific, Rockford, IL, USA) on Aersham HyperfilmTM ECL photographic plates (GE Healthcare, Buckinghamshire, UK).

### 2.8. Detection of ZIKV Viral Elements in EVs from Infected HUVECs

The isolation of EVs of HUVECs was performed under experimental conditions as mentioned above. The EV pellet was resuspended in 200 μL of 1X PBS using the “Total exosome RNA and Protein Isolation kit” (Invitrogen, Vilnius, Lituania), as indicated in the user guide. Subsequently, the isolated RNA was quantified and adjusted to 100 ng per end-point PCR reaction, and the following reactions were carried out: amplification of domain III of the envelope sequence (DIII) using the following primers Forward, 5′ACCCTCGAGTGCGTTCACATTCACC 3′; Reverse, 5′AACGGTACCATACTCCTGTGCCAGTG 3′ with the following amplification conditions: 95 °C 5 min; 35 cycles of 95 °C 30 s, 59 °C 30 s, 72 °C 45 s, 72 °C 5 min, 4 °C. Amplification of the NS2B gene, Forward, 5′ TATAGATCTGAGCTGGCCCCCCTAGC 3′; Reverse, 5′ CGCGGATCCCTACCTTTTTCCAGTCTT 3′ with the following amplification conditions: 95 °C 5 min; 35 cycles of 95 °C 30 s, 59 °C 30 s, 72 °C 45 s, 72 °C 5 min, 4 °C. Amplicons were resolved on a 1% agarose gel in a 1X TBE buffer and analyzed using a photodocumenter (BioRad, Hercules, CA, USA). To determine the viral proteins, the same protocol for Electrophoresis and Western Blot mentioned above was used, using the primary antibodies (Anti-Rabbit-NS1, 1:1000, Genetex, Irvine, CA, USA) (Anti-Rabbit-E, 1:1000, Genetex, Irvine, CA, USA) overnight; as a secondary antibody (Anti-Rabbit-HRP, 1:3000, Thermo Fischer, Rockford, IL, USA) was used for 1 h.

#### Evaluation of the Infectivity of EVs from HUVECs Infected with Zika Virus

Vero cells were seeded at a confluence of 100,000 cells per well on a glass coverslip in a 24-well plate and treated under the following conditions: cells inoculated with 80 ng and 200 ng RNA, respectively, of EVs isolated from HUVECs infected with ZIKV, and the same concentrations of EVs isolated from uninfected HUVECs (Mock); additionally, cells infected at 1 MOI with Zika virus were used as a positive control, and as negative control uninfected cells (mock) were used. The cells were incubation at 37° C with a 5% CO_2_ atmosphere for 24 h. Subsequently, the cells were fixed and prepared for immunofluorescence assay as mentioned above, and the monoclonal antibody 4G2 against Zika virus envelope (The Native Antigen Company, Kidlington, UK), as primary antibody (1:200), was added for 1 h. Anti-mouse Alexa Fluor 488 (Invitrogen, Rockford, IL, USA) (1:500) was added for 1 h. The images were analyzed by fluorescence microscopy.

### 2.9. EV Analysis by Cryo-TEM

The EV fraction was resuspended in sterile 1X PBS and stored at −70 °C. From the isolated exosome fraction, 10 μL per condition was taken. This volume was brought to −170 °C in the semi-automated Cryoplunge–3 kit (Gatan, Pleasanton, CA, USA). Aliquots of 10 μL were placed on a perforated charcoal grid, blotted with filter paper on the grid, and immersed in ethanol −170 °C. The samples were placed in cryoholder 626 (Gatan, Pleasanton, CA, USA), and liquid nitrogen was added, and they were maintained at −180 °C. The sample was mounted on a JEM 2100 LaB6 microscope (JEOL Ltd., Tokyo, Japan); images were captured and analyzed with Digital Micrograph 2.0x software (Gatan, Pleasanton, CA, USA).

### 2.10. EV Analysis by NTA

EV pellets were resuspended in 100 μL of 0.22 µm filtered 1X PBS and stored at −70 °C. For the NTA analysis required for each ZIKV and mock condition, a 1:1000 dilution in a final volume of 1 mL, as well as a Nanosight NS300 equipment (Malvern Panalytical, Malvern, Worcestershire, UK), was used, with a 488 nm laser. NTA 3.2.16 software with a threshold of 5 and gain of 12.6 was used under conditions of a constant flow rate = 50 to 25 °C, a camera level of 12, and 25 frames per second.

#### Statistical Analysis

Unpaired Student’s *t*-test and ordinary one-way ANOVA were performed using GraphPad Prism version 6. *p*-values * < 0.05 and *** *p* < 0.001 were considered statistically significant. The degrees of significance and the test used are indicated in figures: * *p* < 0.05, *** *p* < 0.001, ns = not significant.

## 3. Results

### 3.1. Optimal HUVEC Isolation and Evaluation of Permissibility to ZIKV Infection

Once the HUVECs were isolated, the primary culture was characterized to determine their purity. First, an immunofluorescence assay was performed to identify coagulation Factor VIII, a widely described marker of endothelial cells that plays an essential role in the activation of coagulation [7]. Nuclei stained with DAPI were observed, staining with factor VIII was green, and the overlapping of the images showed 100% double-positive cells (Figure 1A), indicating an optimal purity of the cell culture. A quantitative assay was used to determine the purity percentage, and flow cytometry was performed. First, we analyzed the cellular population according to its size and complexity and subsequently evaluated the markers of anti-CD144/VE–Cadherin and anti-CD31/PECAM. The results showed a 97.9% double-positive population. In addition, the isotype control of the secondary antibodies FITC and PE-Cy7 was evaluated (Figure 1B). These results showed the higher homogeneity and purity of the HUVECs. Finally, we characterized the time at which a higher percentage of cells were infected with ZIKV. Infection kinetics were performed at 16, 24, and 48 hpi at an MOI of 5 and were evaluated by immunofluorescence to identify the ZIKV envelope protein (Figure 1C). In Figure 1D, it can be seen that ZIKV infection does not compromise the viability of the HUVECs; therefore, when evaluating the percentage of infected cells per field, it was found that at 48 hpi, there was the highest percentage of cells infected with ZIKV with an average of a 49% infection (Figure 1E), so this time was determined for the following assays.

### 3.2. ZIKV Infection Changes the Proteome of HUVECs

First, we evaluated whether infection by ZIKV alters the protein profile in HUVECs; the cells were infected at an MOI of 5 for 48 hpi with ZIKV, and infected mock cells were used as a control. Protein extracts were obtained from both experimental conditions and analyzed by Western blotting for evidence of infection and evaluated by 2D electrophoresis to separate proteins by their isoelectric point (pH 3–10) and molecular weight (Figure 2A). The results obtained showed, in mock conditions, the basal proteome of the HUVECs (blue circle); interestingly, under infection conditions, the protein profile showed apparent changes. Several spots with a decrease (green circles) or increase (red circle) are seen in comparison to the basal level in the mock conditions (Figure 2A,B). To further characterize these changes, we performed a qualitative analysis; spots with an apparent decrease and increase in expression were selected, and the densitometric assay determined the presence of 13 proteins that experienced these changes during ZIKV infection; 8 of them were downregulated and 5 were upregulated compared to the control conditions (Figure 2C,D). The first approach to elucidate the identity of the selected spots, an in silico analysis, was performed; for this purpose, the isoelectric point and molecular weights obtained from the 2D assay were considered for each spot, and this information was used to find candidate proteins in the PhosphositePlus database. A summary of the results obtained is shown in Table 1. We were able to characterize three cellular processes that could be regulated during infection with Zika virus in HUVECs; these processes are exosome biogenesis, oxidative stress, and autophagy (Table 1).

### 3.3. EVs Isolated from ZIKV-Infected HUVECs Act as Vehicles for Transporting Infectious Viral Elements

RNA and proteins were isolated from the EVs extract using a commercial exosome-specific kit to determine whether EVs isolated form HUVECs under infectious conditions contained viral elements with infective potential. RT-PCR was performed to detect sequences corresponding to the E protein (domain III) and NS2B regions (Figure 3A,B). Electrophoresis results revealed clear amplification of these sequences in lanes corresponding to EVs from infected cells, with PCR products marching those obtained from infected HUVECs used as positive controls. These amplicons were not detected in EVs derived from infected cells (mock). Additionally, the presence of viral proteins in EVs from infected cells was assessed. Western blot analysis confirmed the presence of the E and NS1 proteins (Figure 3C). Total extracts from infected HUVECs served as positive controls, while these proteins were absent in EVs from uninfected cells. After confirming the presence of viral RNA in the EVs, their infectivity was evaluated. Vero cells were incubated with EVs corresponding to 80 and 200 ng of RNA, and immunofluorescence assays were conducted 24 h post-inoculation. Cells exposed to EVs from infected conditions exhibited a positive signal for the ZIKV E protein, with a higher number of infected cells observed at the higher EV dose (Figure 3D). In contrast, no signal was detected in cells treated with EVs from uninfected cells. Cells infected with ZIKV at an MOI of 1 served as a positive control. These findings suggest that EVs isolated from ZIKV-infected HUVECs carry viral components capable of initiating infection.

### 3.4. Proteins Involved in Exosome Biogenesis Are Altered During Infection of ZIKV-Infected HUVECs

After performing in silico analysis with the possible identities of the proteins regulated during ZIKV infection, we carried out confirmatory assays to prove that these protein candidates were under- or over-regulated by Western blot assays. Protein extracts were obtained from the HUVECs under the aforementioned conditions. Our results showed that ALIX, CD9 (exosomal biogenesis), SOD-1, and CAT (regulation of oxidative stress) had decreased protein expression levels during ZIKV infection, in contrast to the mock-infected cells (Figure 4A). In addition, a densitometric assay confirmed these findings for the ALIX, CD9, SOD-1, and CAT proteins, with significant differences between the infected and mock conditions (Figure 4A). Interestingly, the regulation of exosomal proteins may be related to the hijacking of the biogenic pathway of exosomes during ZIKV infection in HUVECs. Additionally, through infection kinetics at 16, 24, and 48 h, we observed how the regulation of ALIX expression in HUVECs under infection conditions showed a decrease in this protein from 24 h post-infection (Figure 4B), suggesting that infection modulates proteins involved in the biogenic of EVs.

### 3.5. Isolation and Characterization of EVs from HUVECs

We evaluated whether the expression of proteins that regulate exosome biogenesis during ZIKV infection affects the production of these vesicles. Exosomes were isolated using an ultracentrifugation protocol [19]. The exosomal fraction obtained was molecularly analyzed to identify widely described exosomal proteins that serve as markers of these vesicles and a marker that is not present in exosomes. The results showed that the three proteins evaluated (ALIX and CD9 exosome markers, and calnexin, no exosome marker) were present in the total extract of HUVECs (Figure 5A). However, only ALIX and CD9 proteins were detected when assessing the exosome fractions, and no signal was observed for calnexin. Subsequently, flow cytometry was performed to quantitatively evaluate the purity of the isolated exosomes. First, magnetic beads coupled with anti–CD63. These beads have a diameter of 4.5 µm to demonstrate the exosomal complexes. Figure 5B shows a histogram characterizing the population of these complexes of anti–CD63 magnetic beads and exosomes according to their size, independent of cellular debris. This fraction was analyzed using anti-CD9/APC, and the purity of the fraction of exosomes was determined; Figure 5C shows an 85.3% signal for this marker, suggesting that this percentage of vesicles are exosomes. Additionally, to evaluate whether the ultracentrifugation process did not modify the structural integrity of these vesicles, flow cytometry assays were implemented using a CD–9–APC antibody and staining with carboxyfluorescein succinimidyl esterases (CFSE), which only fluoresces when intracellular esterases hydrolyze it, and fluorescence only accumulates in systems that preserve membrane integrity. The results showed a percentage of 53.9% with a double-positive signal for the exosome marker (CD9) and CFSD (Figure 5D), which suggests that these exosomes maintain their integrity during the isolation process.

Finally, a cryo-TEM assay was performed to determine the morphological characteristics of the EVs obtained from HUVECs and to evaluate their physical characteristics such as shape, lipid bilayer presence, and vesicle size. Our results revealed a typical spherical shape with a closed circular lipid bilayer (Figure 5E) in the isolated EVs. Interestingly, an additional morphology was observed that corresponded to EVs with double vesicles. These structures also had their lipid bilayers delimited independently, and the size of each vesicle was different (red squares). Three independent assays were performed, and we found an average of 20 EVs per assay, of which 16 vesicles had a size between 30 and 150 nm. Only four vesicles exceeded this size, ranging between 151 and 200 nm (Figure 5E). Regarding the morphology of the EVs, the average found was 17 EVs, classified as single vesicles and 2 EVs as double vesicles. Taken together, these data indicate that EV isolation by ultracentrifugation is optimal for obtaining a high fraction of these vesicles.

### 3.6. HUVEC EVs During ZIKV Infection Show Differential Regulation in Exosomal Proteins and Quantity and Size of These Vesicles

To corroborate the data obtained from proteomic analysis, exosomes were isolated from ZIKV-infected HUVECs as described in the previous section. From these exosomes, protein extracts were characterized by Western blotting. The total protein extracts from HUVECs infected with ZIKV and mock conditions were evaluated for changes in the protein expression of exosomal markers. Interestingly, using densitometric assays, it was observed that ALIX and CD9 from total extracts of infected cells showed an apparent decrease in these two proteins in comparison with exosome extracts from infected cell isolates, where the expression of ALIX and CD9 increased (Figure 6A). This finding could be related to the enrichment of these proteins and overproduction of these vesicles during ZIKV infection in HUVECs.

We evaluated the hypothesis that ZIKV infection generates an EVs overproduction. The first approach consisted in analyzing the number of EVs using cryo-TEM under the experimental conditions mentioned above. Furthermore, microscopic analysis helped identify the size of these vesicles, showing two populations with a defined range. Under infection conditions, we identified approximately 45 vesicles ranging from 30 to 150 nm, which have been reported as the characteristic size of EVs (Figure 6B). However, we observed another population of 34 vesicles with sizes ranging from 160 nm to 450 nm. Compared to the EVs isolated under mock conditions, a distribution of 16 EVs was observed between 30 and 150 nm, and the remaining six exosomes ranged in size from 153 to 183 nm. Interestingly, under both conditions, the vesicles structurally presented a membrane well delimited by a lipid bilayer, and the morphological changes in the EVs isolated from infected cells presented morphological diversity that is addressed later. Taken together, our data suggest that exosomal proteins are depleted under infection conditions, which is reflected by the enrichment of ALIX and CD9 in the exosomal fraction under ZIKV infection conditions, indicating an increase in the number and size of the EVs in the context of viral infection.

To corroborate and obtain quantitative data on EVs’ size distribution, an NTA assay was performed. The EVs from the mock condition showed a homogeneous size range with only 6 different populations on average, while the EVs from ZIKV-infected HUVECs showed a greater amount of size ranges with 10 detected; the average maximum size detected in the mock was 482 nm, while in the infection conditions it was 656 nm, as shown in the histograms of (Figure 7), and a representative image of each condition is attached, as observed by the field, the number of vesicles detected by NTA. The most remarkable data was determined by the increase in the number of EVs fractions from infected HUVEC; an average concentration of 2.47 × 10^8^ particles/mL was obtained, while the mock EVs had an average concentration of 7.99 × 10^7^ particles/mL. The data from three independent assays were analyzed, and a significant increase in these EVs was observed under infection conditions. Together, these data and the cryo-TEM analysis suggest that under infection conditions, there is a greater distribution of size ranges up to 656 nm on average, and that the infection of cells with ZIKV modulates a greater biogenesis of the number of EVs.

### 3.7. Determination and Alteration of Morphological Populations of EVs During ZIKV Infection in HUVECs

Interestingly, when analyzing the size of EVs by cryo-TEM, different EV morphologies were observed under both mock and ZIKV infection conditions (Figure 8A). The most remarkable finding was that in the mock group, there were only two morphological populations that were classified as single vesicles (Figure 8B) and double vesicles (Figure 8C), but in infection conditions, HUVECs modulated other morphological types that were classified as electrodense vesicles (Figure 8D), triple vesicles (Figure 8E), ruptured membranes (Figure 8F), and double membrane vesicles (Figure 8G). In this sense, it was observed that no other morphological types were present during ZIKV infection, and in comparison, with the mock, single vesicles increased significantly in number as well as the double vesicles; however, this was not the case with double membrane vesicles, which maintained their proportion both in infection and in the mock (Figure 8H). The presence of electrodense vesicles, triple vesicles, and double membrane vesicles opens a new panorama of study on the implications of the presence of these morphological types during ZIKV infection in HUVEC and provides a precedent for the first study of morphological types present in virus infection. Obtaining morphological diversity in the EVs under infection conditions indicates that ZIKV infection not only regulates the size and quantity of the EVs, but also regulates the formation and quantity of each morphological type.

## 4. Discussion

Flavivirus infections are relevant because of their impact on health and ease of spread. Viral pathogenesis can be studied using 2D proteomic assays, which analyze alterations in the expression of proteins directly regulated by viruses. In these assays, protein expression decreased during DENV infection, and the expression of the PHB1 protein decreased. This alteration is related to the success of the infection and its association with viral protein E [25]. Similar alterations have been reported during CHIKV (arbovirus) infection, in which the underexpression of proteins involved in several interrelated cellular pathways has been observed, including cell signaling, lipid metabolism, protein modification, transcription, translation, and stress responses [26].

Our 2D proteomic analysis revealed several proteins with over- or underexpression during ZIKV infection; the most representative proteins were selected for densitometric analysis (Figure 2). Using an in silico assay, we analyzed the molecular weight and isoelectric point of each spot and compared them with the phosphosite database (Table 1). We identified the proteins involved in exosome biogenesis and loading (ALIX and CD9), oxidative stress (Catalase and SOD-1), and autophagy (Beclin-1). These results were confirmed by Western blotting of protein extracts from ZIKV-infected HUVECs, focusing on ALIX, CD9, Catalase, and SOD-1 proteins (Figure 4). Previous studies have shown that SOD-1 and Catalase are overexpressed in trophoblasts as regulatory mechanisms against reactive oxygen species (ROS) [27].

Interestingly, an increase in ROS has been described as being shared among Flaviviridae [28]. Our data are consistent with those observed in ZIKV-infected neuronal and liver cells, in which catalase and SOD-1 levels decrease during infection, facilitating viral pathogenesis [29]. The low expression of catalase and SOD-1 in ZIKV-infected HUVECs offers a new perspective on the regulation of ROS in this cellular environment. Furthermore, the regulation of crucial proteins involved in exosome biogenesis, loading, and release, which are essential for maintaining cellular homeostasis during infection, was analyzed.

However, the finding about the regulation of protein involved exosomes biogenesis during ZIKV infection, giving an idea in relation to the possibility that these vesicles are regulated during infection. In this study, different methodologies were employed to characterize vesicle isolation [30,31]. Western blot assays with several markers showed the presence of ALIX and CD9 in the exosome fraction, and the absence of calnexin protein in these fractions indicated that the isolation process was successful. Additionally, flow cytometry assays revealed an 85% signal-positive marker, CD63, which is similar to that reported in another study of these vesicles 75–82% signal-positive [32].

Our results also showed that the integrity of the vesicle was preserved during the isolation process, with a 53.9% double-positive signal observed between CFSE and CD9 markers, demonstrating that the vesicle maintained its integrity and did not rupture. Our data are optimal because, when compared with other studies, 15–32% integrity was observed using the same methodology [33]. These data suggest that intact exosomes perform better at retaining biomolecular cargo.

In addition, cryo-TEM analysis was performed in order to examine the bilipidic and spherical structure of the EVs [18]. EVs with a size between 30 and 150 nm were identified (Figure 5E), and other morphological structures with double vesicles were observed. There is no consensus regarding their classification in the morphological reports of EVs. Nevertheless, the authors agree that single-, double-, multivesicular, and electron-dense vesicles contain a significant number of proteins and lipids as well as double and broken membranes. These morphological populations have been found in the semen, human follicles, cerebrospinal fluid, plasma, breast milk, HMEC-1, THP-1, MDA321, and GT1-7 cells [20,21,22,34]. Cryo-TEM analysis revealed that 84.2% of EVs were 30–150 nm in size and 15.7% were 151–200 nm, consistent with studies establishing vesicular sizes as large as 180 nm [35,36].

After the isolation and characterization of exosomes, our results showed an increase in ALIX and CD9 proteins in the exosome fraction during ZIKV infection, with differences in total protein where these proteins were downregulated (Figure 6A), suggesting a depletion that could increase exosome production during ZIKV infection. In addition, the results obtained by cryo-TEM and NTA confirmed this idea to identify the amount and size of vesicles in ZIKV conditions, and it was observed that there was a greater amount of vesicles (2.47 × 10^8^ particles/mL) under the infection conditions, and their size showed heterogeneity as a size range of 30 to 600 nm; these findings confirm that ZIKV can modulate exosome biogenesis and production in HUVECs. Further studies are required to explore the composition of these vesicles and their roles in ZIKV pathogenicity.

Furthermore, our findings indicate that EVs isolated from ZIKV-infected HUVECs contain viral components that may contribute to infection spread or cellular dysfunction. Specifically, the detection of E and NS1 viral proteins in these EVs is consistent with previous studies on cellular models of the maternal–fetal interface [37,38]. Although the precise effects of these proteins when delivered via EVs to recipient cells remain to be elucidated, their presence suggests a potential role in modulating membrane permeability or triggering immune activation, thereby contributing to tissue damage. Moreover, we identified viral RNA genomes within EVs, in agreement with previous reports in other models of ZIKV infection. Notably, these genome-containing EVs were capable of initiating infection in recipient cells, underscoring their potential role as mediators of viral dissemination. This raises the possibility of an alternative route across the EVs for transplacental transmission of ZIKV, which could facilitate viral spread to various fetal tissues and organs.

Our results are in agreement with a research work, who reported a decrease in ALIX and CD9 proteins in ZIKV-infected HUVEC-CLR-1730 cells, whereas in the exosomal fraction, both proteins showed an increase in expression during infection, similar to what was observed in this study [37]. However, our results were based on primary cultures of umbilical vein endothelial cells, which provided a more representative study model. Furthermore, the hijacking of the exosomal pathway by viruses affects the regulation of exosomal biogenic proteins. However, studying the biogenic proteins of exosomes in viral infections is crucial because tetraspanins, such as CD81, CD63, and CD9, are involved in viral release and regulate the increase in exosome production [39,40,41]. Interestingly, the cryo-TEM analysis of EVs infected with ZIKV revealed changes in morphological diversity. In this study, the morphological types of EVs were classified as single, double, and triple vesicles; electron-dense vesicles; and those with double or broken membranes (Figure 8A–G).

There are disparities in the morphological classification of EVs, and their morphological diversity has been observed to be altered in non-infectious diseases. The EVs in the plasma of patients with breast cancers were classified into five morphological types: single, double, multivesicular, electron-dense, and double-membrane vesicles. The proportions changed when comparing the vesicle types between healthy individuals and cancer patients. In the whole blood of healthy individuals, the proportions of each type were more equal, with 37% of single vesicles; however, in patients with breast cancer, this percentage increased to 70%. Furthermore, although electron-dense vesicles were not observed in the plasma of healthy individuals, they were observed in patients with breast cancer [42].

In Parkinson’s disease, EVs have been isolated from the cerebrospinal fluid, and cryo-TEM analysis has shown that patients present different proportions of morphological types. Patients show fewer single vesicles than healthy individuals, whereas multivesicular vesicles are less frequent in healthy individuals [43]. A study on infectious prion disease also analyzed morphological types; in murine GT1-7 cells infected with the M1000 prion, three types of vesicles were identified: single, double, and multivesicular. During infection, the number of single vesicles increases and multivesicular vesicles decrease [24].

The regulation of diverse EV morphologies remains unclear; however, some hypotheses point to BAR family proteins. With their positively charged crescent shape, these proteins interact with negatively charged membrane lipids, forming filaments that stabilize membrane curvature. For example, F-BAR stabilizes endocytic invagination and regulates vesicle expansion, whereas I-BAR induces negative curvatures in the membrane [44,45,46]. Another possible regulatory mechanism is found in lipids during ZIKV infection, since ceramide, sphingomyelin, phosphatidylethanolamine, and cholesterol in the lipid bilayer have geometric shapes that facilitate diverse curvatures [47,48].

Proteomic analysis using 2D electrophoresis presents limitations in terms of the breadth of detectable protein candidates, especially when compared to next-generation technologies such as tandem MS, LC-MS/MS, and LC-MS. These advanced techniques allow for more comprehensive and detailed protein identification, as well as the analysis of complex protein interaction networks. In this regard, the 2D proteomic approach provides a valuable initial insight into the proteomic map of HUVECs under infection conditions. This method enables the identification of potentially dysregulated proteins during infectious events, serving as a starting point for more in-depth studies using higher-resolution technologies.

Overall, evidence suggests that ZIKV may regulate or hijack exosome biogenesis. This study provides the first insight into this phenomenon, as the morphological populations observed in ZIKV-infected HUVECs show similarities to the morphological types reported in non-infectious disease studies.

## 5. Conclusions

The hijacking of the exosomal pathway by ZIKV promotes an increase in EVs and regulates the expression of proteins involved in exosomal biogenesis. HUVECs present under infection conditions and different morphological populations of EVs suffer an alteration in terms of quantity and morphological types by ZIKV infection. Furthermore, these EVs carry viral elements that may contribute to the dissemination of infection. This opens a new panorama of studies to understand the implications of each morphological type during ZIKV infection in HUVECs and, more importantly, the cellular and viral factors present in each morphological type.

## Figures and Tables

**Figure 1 microorganisms-13-01402-f001:**
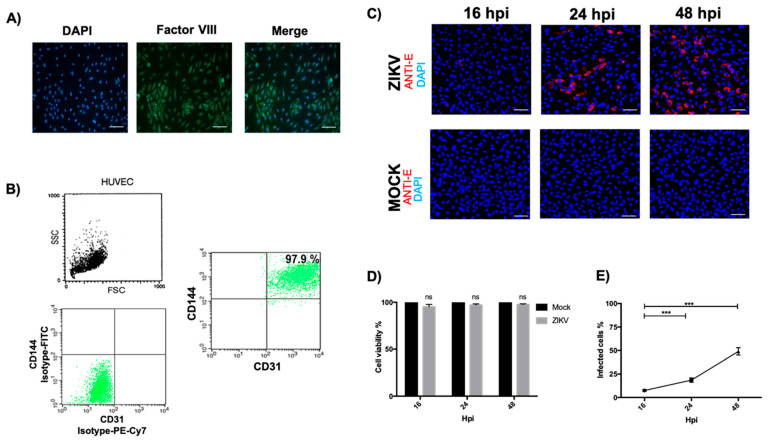
Infection of umbilical cord endothelial cells with ZIKV. (**A**) Immunofluorescence (IF) of HUVECs was performed showing staining, with DAPI showing blue nuclei as a marker of HUVECs, which was used in Factor VIII coagulation showing green staining; the splicing of both images is shown (magnification 20×). (**B**) The percentage of HUVEC isolation was evaluated by flow cytometry, determining the HUVEC population by size and complexity. Additionally, the isotype control is observed for FITC and PE-Cy7 antibodies; CD31 and CD144 markers were used to show a double positive and determination of cell lineage purity. (**C**) The maximum percentage of cells positive for anti-ZIKV envelope was determined by IF with infection kinetics at 16, 24, and 48 hpi at an MOI of 5 showing red staining, while nuclei in blue were stained with DAPI (40× magnification). (**D**) Evaluation of the percentage of viability of HUVECs infected with ZIKV. (**E**) Analysis of the percentage of positive cells per visual field for ZIKV infection. All experiments were performed independently by 3 donors. Student’s *t*: ns, not significant; ANOVA, *** *p* < 0.001; scale bars, 20 μm.

**Figure 2 microorganisms-13-01402-f002:**
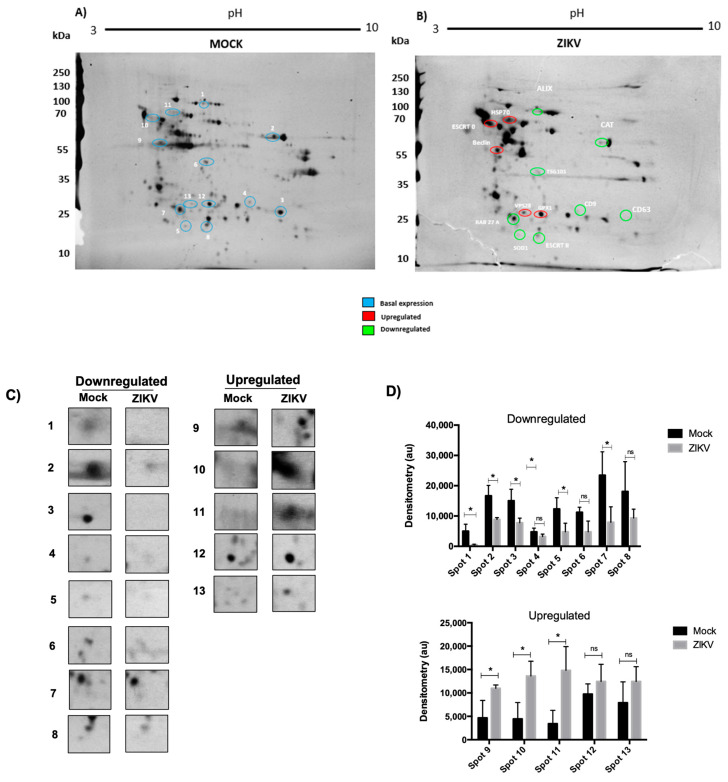
ZIKV infection alters the proteome of HUVECs. Total extracts were obtained from infected HUVECs at 48 hpi at 5 MOI, and two-dimensional gels were run. Proteins were separated by their isoelectric point (3–10 pH) and by their molecular weight showing a 250 kDa molecular weight marker. (**A**) Spots are compared with a gel of basal mock extracts (left). (**B**) The blue color shows the spots in basal state, selected in the gel under infection conditions (right), which indicates the spot’s overexpression in red while the green color indicates underexpression. (**C**) The most representative spots with overexpression and underexpression are selected and zoomed for analysis. (**D**) Densitometric analysis of the selected spots compared with the control or basal gel (mock). The gels were compared with each other; the fold change values as well as the *p* values of the selected spots were calculated in Image J 1.47v software using analysis. All experiments were performed independently by 3 donors. Spots with Student’s *t* * *p* value < 0.05 and an absolute abundance change (fold change) > 1.5 were considered differential; ns, not significant.

**Figure 3 microorganisms-13-01402-f003:**
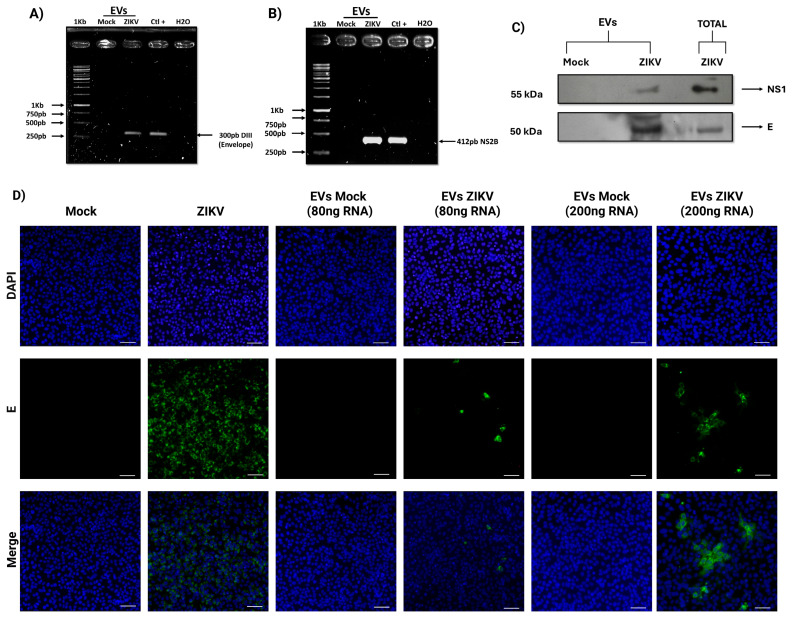
Analysis of viral elements contained in extracellular vesicles isolated from ZIKV-infected HUVECs and their infectivity. (**A**,**B**) RT-PCR of EVs RNA: (**A**) amplification of the viral envelope DIII sequence (300 bp). (**B**) Amplification of the viral NS2B sequence (412 bp). Controls: Mock (EVs from uninfected cells), no amplification products; Ctl+ (total RNA from infected cells) with an amplification product 300 bp and 412 bp; H_2_O (no template) no amplification products, ruling out contamination. (**C**) Western blot: Detection of NS1 (55 kDa) and E (50 kDa) viral proteins in total extracts and EVs of infected cells; mock uninfected cells. (**D**) Effect of extracellular vesicles (EVs) on Vero cells (magnification 10×). Cell nucleus were stained with DAPI (blue). ZIKV infection was detected using an antibody against the viral envelope protein (green). Mock cells and cells treated with EVs from uninfected cells (10 and 30 µL) showed no infection. Cells directly infected with ZIKV (1 MOI) showed 90% infection. Cells treated with 10 µL (80 ng RNA) of infected EVs showed <5% infection. Cells treated with 30 µL (200 ng RNA) of infected EVs showed 15% of infected cells.

**Figure 4 microorganisms-13-01402-f004:**
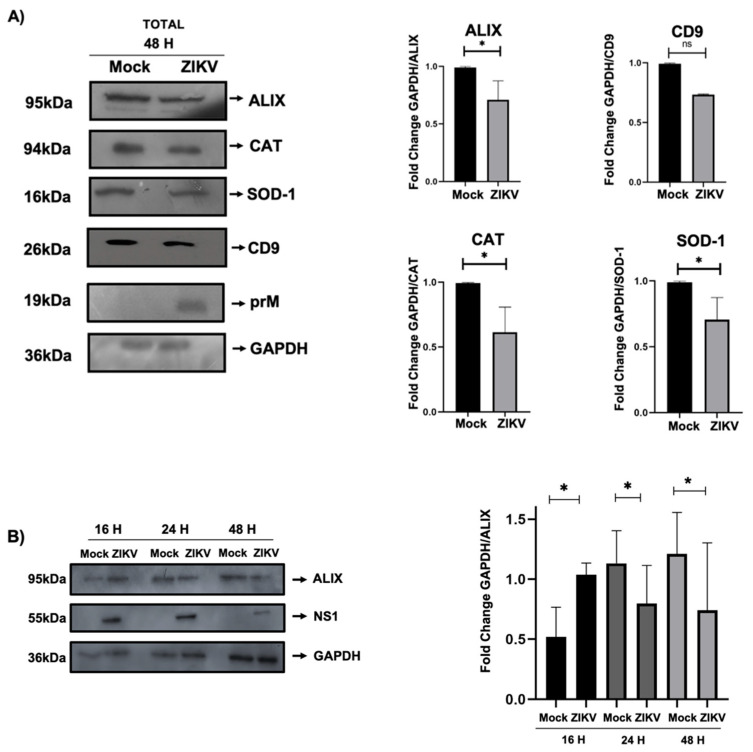
ZIKV infection alters HUVEC proteins. (**A**) Total lysates were obtained from ZIKV-infected HUVECs to confirm that the selected spots, together with their identity by in silico analysis, show underexpression. The proteins evaluated were ALIX, CD9, Catalase (CAT), and SOD1, with prM as infection control and GAPDH as loading control. Densitometric analysis showed that ALIX, CAT, and SOD–1 showed a significant decrease; CD9 protein expression was not significant. (**B**) Analysis of total lysates of HUVECs infected by ZIKV during an infection kinetic at 16, 24, and 48 h; 5 MOI; observing the expression of ALIX protein involved in the biogenesis of exosomes. NS1 was used as a control of ZIKV infection and GAPDH as a loading control. A densitometric analysis was performed, revealing a decrease in ALIX after 24 h post-infection. All experiments were performed independently by 3 donors; Student’s *t*; * *p* < 0.05; ns, not significant.

**Figure 5 microorganisms-13-01402-f005:**
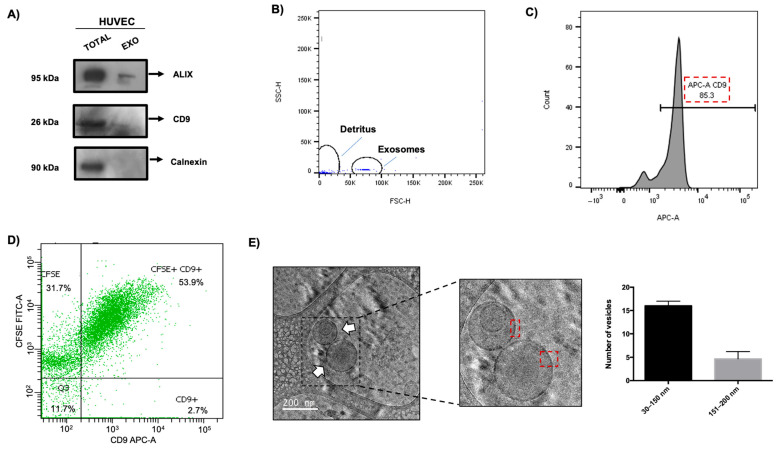
Isolation and characterization of exosomes from HUVECs. (**A**) Protein expression of ALIX, CD9, and Calnexin as a negative marker of exosomes; extracts of HUVECs (total) were used as a positive control, and extracts of the exosome fraction (exo) secreted by these same cells were analyzed by Western blot. (**B**,**C**) Analysis of exosome purity by flow cytometry, graph delimiting the population of exosomes coupled to the anti-CD63 magnetic beads by complexity and size. The percentage of isolate by magnetic beads with CD9 staining present on exosomes from HUVECs. (**D**) Integrity of HUVEC exosome isolate. The graph shows 85.3% double positive for both CFSE and CD9, according to the integrity of these vesicles by CFSE and the positive signal for CD9 belonging to a tetraspanin present in the exosomes. The CFSE-only population was undetermined. (**E**) Characterization of EVs isolated from HUVECs by cryo-TEM; the red box indicates a lipid bilayer. Nineteen vesicles are visualized, of which sixteen maintain a size of 30–150 nm, while four vesicles are 151–200 nm in size. The morphology of HUVEC EVs shows only 2 varieties: single vesicles (17) and double vesicles (2). Representative images obtained by cryo–TEM from 3 independent assays are presented; white arrows show EVs.

**Figure 6 microorganisms-13-01402-f006:**
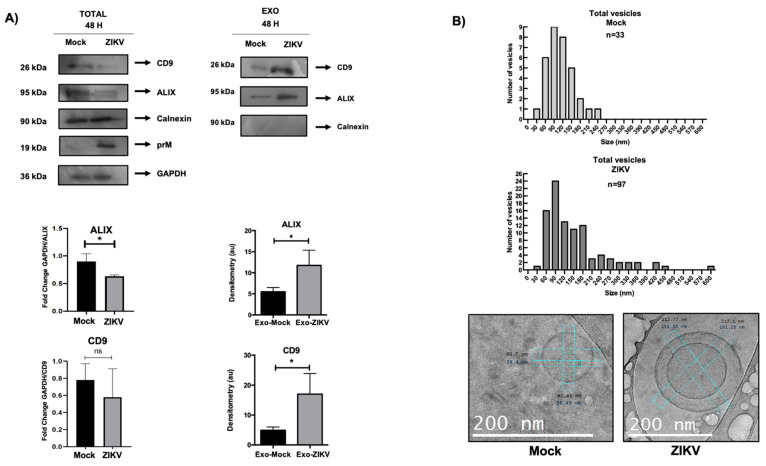
ZIKV infection increases the amount and size of EVs produced by HUVECs. (**A**) Analysis of protein extracts of exosomes under infection conditions. Exosomal markers were evaluated by Western blot of total extracts of HUVECs (Total) and exosome extracts of HUVECs (Exo). Both extracts were obtained under ZIKV infection conditions at 48 h and 5 MOI. The exosome markers used were CD9, ALIX, and Calnexin as negative control, while the infection marker ZIKV prM and GAPDH was used as loading control. Characterization of exosomes under ZIKV and mock infection conditions. (**B**) Cryo-TEM assay of isolated EVs under infection conditions. Measurements of EVs under infection and mock conditions were performed by Digital Micrograph 2.0X software. Under mock conditions, the isolated EVs showed homogeneity in terms of size distribution analyzed with 23 vesicles. The graph shows the heterogeneity in size distribution of exosomes isolated from ZIKV-infected HUVECs, with 79 vesicles found. The images captured by cryo-TEM show the characteristic morphology of the EVs under both ZIKV infection and mock conditions. Student’s *t*: * *p* < 0.05; ns, not significant.

**Figure 7 microorganisms-13-01402-f007:**
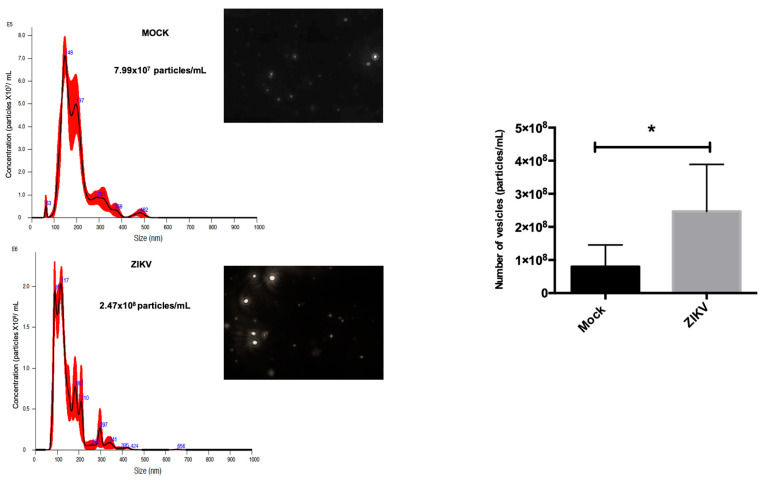
Increased number of EVs during ZIKV infection in HUVECs. (A) On average 2.47 × 10^8^ particles/mL were obtained in EVs isolated from the infection; on the other hand, in the mock 7.99 × 10^7^ particles/mL, two representative images of the NTA are observed, in red shows the distribution of three consecutive runs of each sample. All experiments were performed independently by 3 donors; Student *t*, * *p* < 0.05.

**Figure 8 microorganisms-13-01402-f008:**
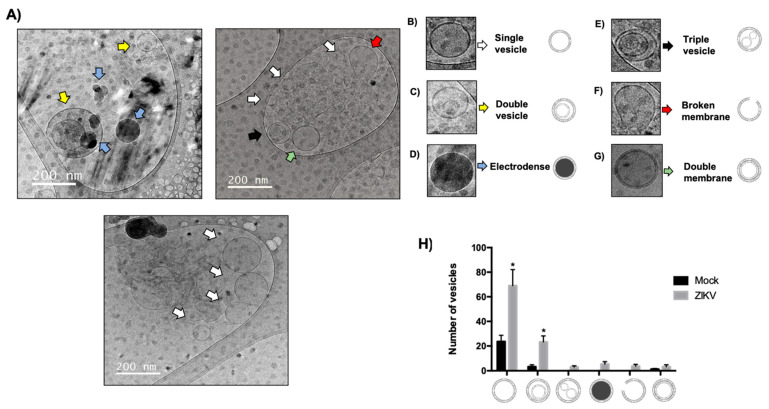
ZIKV infection changes the morphology of EVs produced by HUVECs. Cryo-TEM morphological analysis of EVs isolated from ZIKV-infected HUVECs. A morphological variety of EVs were observed under infection conditions. (**A**) Shown in white arrows are single vesicles (**B**), yellow indicates double vesicles (**C**), blue arrows electrodense vesicles (**D**), black arrows triple vesicles (**E**), red arrows broken membrane vesicles (**F**) and double membrane vesicles (**G**). (**H**) A histogram is shown analyzing the morphological types present in conditions of ZIKV infection; independent triplicates were performed. Student *t*, * *p* < 0.05.

**Table 1 microorganisms-13-01402-t001:** In silico analysis of differentially expressed proteins between mock conditions and ZIKV infection.

Spots	Protein	MW	pI	Funtion	Regulation During Infection
**1**	**ALIX**	**96.02**	**6.13**	**Protein cargo, biogenesis exosomes**	**Down**
**2**	**Catalase**	**59.75**	**6.9**	**Oxidative stress regulator**	**Down**
3	CD63	25.637	8.14	Tetraspanin exosomes	Down
**4**	**CD9**	**25.41**	**6.8**	**Tetraspanin exosomes**	**Down**
**5**	**SOD-1**	**15.93**	**5.7**	**Oxidative stress regulator**	**Down**
6	TSG101	43.9	6.06	ESCRT-I interacting protein, for sorting	Down
**7**	RAB27A	24.86	5.09	GTPase leads to MVBs	Down
8	ESCRT II (VPS 25)	20.74	5.97	Exosome biogenesis	Down
9	BECLIN-1	51.896	4.83	Precursor of autophagy	Up
10	STAM (ESCRT 0)	59.18	4.7	Exosome biogenesis	Up
11	HSP70	70.03	5.5	Heat shock protein and protein exosome	Up
12	GPX1	22.08	6.15	Oxidative stress regulator	Up
13	VPS28	25.42	5.37	ESCRT-I regulatory protein	Up

## Data Availability

The original contributions presented in this study are included in the article. Further inquiries can be directed to the corresponding author.

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
