# Peer review of "Zika Virus Infection Modulates Extracellular Vesicle Biogenesis and Morphology in Human Umbilical Cord Endothelial Cells: A Proteomic and Microscopic Analysis"

_microorganisms, 2025, doi:10.3390/microorganisms13061402_

Round 1
Reviewer 1 Report
Comments and Suggestions for Authors
This study investigates how Zika virus (ZIKV) infection affects extracellular vesicle (EV) biogenesis and morphology in human umbilical vein endothelial cells (HUVECs), using proteomic analysis (2D electrophoresis, Western blot) and advanced microscopy techniques (Cryo-TEM, NTA) for characterization. The findings reveal that ZIKV infection increases EV production, alters their morphology, and modifies the expression of key EV-associated proteins (such as ALIX and CD9), potentially influencing viral transmission and maternal-fetal interface functions.
- The study does not explicitly determine whether EVs carry ZIKV RNA or proteins, making it unclear if EVs play a role in viral transmission. It is recommended to perform qRT-PCR to detect ZIKV RNA in EVs, confirming the presence of viral genomes, and Western blot to assess whether EVs contain ZIKV structural proteins (Env, NS1, NS5), evaluating their potential for viral dissemination.
- It is suggested to infect HUVECs with ZIKV, collect EVs, and transfer them to uninfected HUVECs or human neural progenitor cells (hNPCs)to assess whether ZIKV can spread via EVs. Viral replication in recipient cells should be evaluated using plaque assays or qRT-PCR.
- While HUVECs represent placental vascular endothelium, the study does not examine whether trophoblast cells (such as BeWo and JEG-3) exhibit similar EV alterations. Expanding experiments to trophoblast cell lines (BeWo, JEG-3)and comparing EV changes could provide further insights into maternal-fetal transmission mechanisms.
- A comprehensive proteomic analysis using LC-MS/MSshould be conducted to identify systemic changes in EV protein expression post-infection and explore key regulatory factors.
- ZIKV-infected EVs should be used to treat immune cells (macrophages, T cells)to assess whether they influence cytokine secretion (ELISA) or immune activation (flow cytometry analysis of CD80/CD86 expression)
Author Response
30 May 2025
Dear Reviewers
I am pleased to resubmit for publication the revised version of “Zika virus infection modulate extracellular vesicles biogenesis and morphology in human umbilical cord endothelial cells: A proteimic and microscopic analysis ”. I appreciated the constructive criticism from the associated editor and reviewers. I have addressed each of their concerns as outlined below.
Following the reviewer’s advice, I, along with my collaborators have been carefully revised and appropriate changes have been made in accordance with the reviewer’s suggestions. The responses to their comments are provided below:
We appreciate the recommendation and have contacted a service for correction of style and writing in English, through which the manuscript has been polished. In addition, a certificate of the company we hired will be attached.
Reviewer 1:
1.-The study does not explicitly determine whether EVs carry ZIKV RNA or proteins, making it unclear if EVs play a role in viral transmission. It is recommended to perform qRT-PCR to detect ZIKV RNA in EVs, confirming the presence of viral genomes, and Western blot to assess whether EVs contain ZIKV structural proteins (Env, NS1, NS5), evaluating their potential for viral dissemination.
We consider relevant the observation to reviewer, and basically, we performed the assay to evaluate whether EVs harvested under infection conditions contained viral elements such as genomes and proteins, we conducted different assays. The results demonstrate that these EVs indeed carry viral RNA, as well as the E and NS1 proteins. These findings are consistent with reports from other cellular models infected with Zika virus. In the revised version of the manuscript, the corresponding updates are highlighted in blue. However, we believe that a novel and noteworthy aspect of our study is the observation that ZIKV infection alters the morphology of EVs. For this reason, we have strengthened our manuscript by including experimental data supporting the presence of viral components in EVs.
2.-It is suggested to infect HUVECs with ZIKV, collect EVs, and transfer them to uninfected HUVECs or human neural progenitor cells (hNPCs)to assess whether ZIKV can spread via EVs. Viral replication in recipient cells should be evaluated using plaque assays or qRT-PCR.
We appreciate the observation, and we considered it pertinent to perform the infectivity assay once we confirmed that these EVs contained viral genomes. Therefore, we proceeded with experiments to evaluate whether the EVs were infectivity. Although we initially planned to use a neuronal cell model, however the cell line obtained from another researcher at our institute was positive for mycoplasma contamination, and we were unable to continue using it. Consequently, we conducted the assay using the Vero cell line, a widely accepted model for studying various viral infections. The results demonstrated that EVs derived from ZIKV-infected HUVECs were indeed infectious, as evidenced by positive immunofluorescence staining for the viral E protein 24 hours post-inoculation. We had also planned to quantify viral RNA and perform PFU assays; however, due to a failure in our institute’s ultra-freezing system, critical reagents, virus stocks, and cell lines were compromised. The corresponding revisions to this section have been underlined in blue in the updated manuscript.
3.- While HUVECs represent placental vascular endothelium, the study does not examine whether trophoblast cells (such as BeWo and JEG-3) exhibit similar EV alterations. Expanding experiments to trophoblast cell lines (BeWo, JEG-3) and comparing EV changes could provide further insights into maternal-fetal transmission mechanisms.
We appreciate your comment. At the structural level of the maternal–fetal interface, the endothelium of the fetal capillaries are branches that ultimately connect to the umbilical cord, for this reason is considered the final barrier separating fetal circulation from the maternal environment. Therefore, research using these cellular models is essential to understanding the mechanisms Zika virus employs to access fetal circulation. However, the reviewer is correct in pointing out the importance of evaluating whether this phenomenon of EVs morphological changes is also observed in other placental cell types, such as trophoblasts. Our group has access to these cell lines, and we recognize that JEG-3 is the most suitable model for studying ZIKV infection. In contrast, BeWo cells, which can differentiate into syncytiotrophoblasts, exhibit very low permissiveness to ZIKV. Although we have successfully isolated EVs from JEG-3 cells, we have not yet been able to perform microscopy-based analyses. It is also important to note that, JEG-3 cell line is derived from a choriocarcinoma; and our model used in this study involves primary cultures. Therefore, we believe the data obtained from our study offer a more physiologically relevant perspective on this phenomenon.
.
4.- A comprehensive proteomic analysis using LC-MS/MSshould be conducted to identify systemic changes in EV protein expression post-infection and explore key regulatory factors.
We appreciate your valuable comment. We agree that this aspect is essential to identifying which proteins may be present in EVs and how they might regulate signaling pathways in recipient cells during infection. Our group has begun to explore this question, and preliminary data suggest that EV content under infectious conditions is enriched in proteins involved in exosome biogenesis, lipid metabolism regulation, and viral entry mechanisms. Interestingly, our initial analysis also identified proteins related to pathways that control neurodevelopment. While these findings are preliminary, we recognize the need to perform replicates and validation experiments to confirm them. We are currently seeking funding to complete this study.
5.-ZIKV-infected EVs should be used to treat immune cells (macrophages, T cells) to assess whether they influence cytokine secretion (ELISA) or immune activation (flow cytometry analysis of CD80/CD86 expression).
We appreciate your observation and consider that our main contribution remains the description of the morphological changes in EVs induced by ZIKV infection. In this context, we are interested in continuing to investigate these structures. To this end, we are currently conducting a lipidomic analysis of EVs to explore cellular lipid metabolism pathways that may influence the composition of these biomolecules in EVs, which could in turn affect their morphology. We believe that the suggestion you provided could be of interest for a future study conducted by our group.
Thank you for your comment. We have addressed it and made the corresponding changes, which can be seen highlighted in yellow in the revised version of the manuscript.
Finally, we again thank you for your suggestions and insights, which have enriched the manuscript and produced a more balanced and better account of the review. We hope that the revised manuscript is now suitable for publication in the prestigious journal that you represent.
I look forward to your reply.
Sincerely,
Dr. León-Juárez Moisés
Departamento de Inmunobioquímica
Instituto Nacional de Perinatología ‘Isidro Espinosa de los Reyes’
Montes Urales #800, Col. Lomas de Virreyes
CP 11000, Ciudad de México, México.
01 55 5520 9900 ext-438
E-mail: moisesleoninper@gmail.com

Reviewer 2 Report
Comments and Suggestions for Authors
This well-written and clear study explores the isolation and characterization of human umbilical cord vein endothelial cells (HUVEC) and their infection with Zika virus (ZIKV). The main findings highlight the establishment of an effective in vitro model, demonstrating significant insights into ZIKV's impact on endothelial cells and the exosome pathway. Key observations include the identification of specific exosomal proteins and changes in vesicle morphology during viral infection. Overall, the study provides valuable information for understanding ZIKV pathogenesis and its interaction with host cells. I have only few comments.
- Please avoid mentioning studies in the abstract section, as references can not be included.
- Conclusion and perspectives must be included in the abstract section.
- Use keywords different from the title.
- The introduction is comprehensive and well-detailed, but it could be made more concise and focused.
- Ensure that all statements are properly supported by citations.
- Specify the dilution and incubation times for all primary and secondary antibodies.
- Mention the settings and parameters used on the confocal microscope (e.g., laser power, acquisition time).
- Figure 2 A. Please remove the red below KDa. Where is the ladder? Why did the authors remove it?
Author Response
30 May 2025
Dear Reviewers
I am pleased to resubmit for publication the revised version of “Zika virus infection modulate extracellular vesicles biogenesis and morphology in human umbilical cord endothelial cells: A proteimic and microscopic analysis ”. I appreciated the constructive criticism from the associated editor and reviewers. I have addressed each of their concerns as outlined below.
Following the reviewer’s advice, I, along with my collaborators have been carefully revised and appropriate changes have been made in accordance with the reviewer’s suggestions. The responses to their comments are provided below:
We appreciate the recommendation and have contacted a service for correction of style and writing in English, through which the manuscript has been polished. In addition, a certificate of the company we hired will be attached.
Reviewer 2
1.- Please avoid mentioning studies in the abstract section, as references can not be included.
We appreciate the observation of the reviewer and we have made the changes related to this comment.
2.- Conclusion and perspectives must be included in the abstract section
We appreciate the observation of the reviewer and we have made the changes related to this comment. We added this paragraph:In conclusion, our data suggest that ZIKV infection can modulate cellular factors involved in formation and morphology of EVs in HUVEC cells. Furthermore, these EVs carry viral elements that may contribute to the dissemination of infection. Futures studies aimed at analysis proteomic and lipidomic composition of these EVs are needed to understand the biological implications in vertical infection.
3.- Use keywords different from the title.
We appreciate the observation of the reviewer and we have made the changes related to this comment.
4.- The introduction is comprehensive and well-detailed, but it could be made more concise and focused
We have made the changes related to this comment.
5.- Ensure that all statements are properly supported by citations
We appreciate the observation of the reviewer and we have made the changes related to this comment.
6.- Specify the dilution and incubation times for all primary and secondary antibodies.
Thank you for your comment. We have addressed it and made the corresponding changes, which can be seen highlighted in yellow in the revised version of the manuscript.
7.- Mention the settings and parameters used on the confocal microscope (e.g., laser power, acquisition time
Thank you for your comment. We have addressed it and made the corresponding changes, which can be seen highlighted in yellow in the revised version of the manuscript.
8.-Figure 2 A. Please remove the red below KDa. Where is the ladder? Why did the authors remove it?
We appreciate the observation of the reviewer and we have made the changes related to this comment.
Thank you for your comment. We have addressed it and made the corresponding changes, which can be seen highlighted in yellow in the revised version of the manuscript.
Finally, we again thank you for your suggestions and insights, which have enriched the manuscript and produced a more balanced and better account of the review. We hope that the revised manuscript is now suitable for publication in the prestigious journal that you represent.
I look forward to your reply.
Sincerely,
Dr. León-Juárez Moisés
Departamento de Inmunobioquímica
Instituto Nacional de Perinatología ‘Isidro Espinosa de los Reyes’
Montes Urales #800, Col. Lomas de Virreyes
CP 11000, Ciudad de México, México.
01 55 5520 9900 ext-438
E-mail: moisesleoninper@gmail.com

Reviewer 3 Report
Comments and Suggestions for Authors
Comments
The work of Velásquez-Cervantes, M. et al. describes the characterization of exosomes from HUVEC. The idea is very interesting and the work well performed. However, currently, there are more sophisticated methods to analyses the protein content of cells. The introduction is clear. Some references are missing. The methods are, in most cases, ok; however, some parts need further details. Results are well presented. The discussion has all the necessary information and the conclusions are well supported by the results. Define the acronyms used throughout the manuscript.
Major comments
- Some important typos throughout the manuscript.
- What is the novelty in the work?
- Which are the advantages of your approach compared to the state-of-the-art proteomic analysis?
Minor comments
Line 49 – “(…) condition. Interestingly, (…)”
Line 55 – The reviewer suggests adding Western Blot as a keyword since the proteomics is performed using this technique.
Line 75 – “(…) extracellular vesicles (EVs), currently 75 markers are used to discriminate (…)” to “(…) vesicles (EVs). Currently (…)”.
Line 75 – 84 and line 85 – 93 – The references only in the end does not seems correct. Use them along the paragraph.
Line 75 – Is the work on exosomes or EVs in general?
Line 131 “(…) (INPer), the project (…)” to “(…) (INPer). The project (…)”
Line 145 – Add where the Vero Cells came from.
Line 164 – Objective and lasers used.
Line 165 – Is there any special FBS used to isolate exosomes, such as the use of a FBS exosomes-free?
Line 174 – Any quantification on the amount obtained?
Line 176 – What is a “cell package”?
Line 178 – hour is h not H
Line 176 – Add the origin of the antibodies and the reactivity (human, goat, mouse, rat,…)
Line 185 – How many events were acquired?
Line 256 – HUVEC
Line 297 – Infected mock cells were used as a control? Or cells not infected?
Line 574 – Is this the novelty of this work? A proteomic analysis was never performed in HUVEC cells infected with ZIKV?
Line 608—The reviewer suggests adding a paragraph outlining the study's limitations. Nowadays, proteomics relies on very sophisticated methodologies involving LCMS, which are more precise.
Comments on the Quality of English LanguageA lot of typos and a lack of proper punctuation.
Author Response
30 May 2025
Dear Reviewers
I am pleased to resubmit for publication the revised version of “Zika virus infection modulate extracellular vesicles biogenesis and morphology in human umbilical cord endothelial cells: A proteimic and microscopic analysis ”. I appreciated the constructive criticism from the associated editor and reviewers. I have addressed each of their concerns as outlined below.
Following the reviewer’s advice, I, along with my collaborators have been carefully revised and appropriate changes have been made in accordance with the reviewer’s suggestions. The responses to their comments are provided below:
We appreciate the recommendation and have contacted a service for correction of style and writing in English, through which the manuscript has been polished. In addition, a certificate of the company we hired will be attached.
Reviewer 3
1.-Some important typos throughout the manuscript
We appreciate the observation of the reviewer and we have made the changes related to this comment.
2.-What is the novelty in the work?
Our study offers insight into a phenomenon that has been scarcely described in the field of extracellular vesicles (EVs): the modulation of EV morphology under conditions such as viral infections. It is well established that EVs can be influenced by viruses to enhance the dissemination of infection, as viral components, including genomes, can be encapsulated within them. We have strengthened this version of the manuscript by demonstrating that: Zika virus infection in HUVEC cells regulates the biogenesis of EVs and mobilizes viral genomes into them, enabling these vesicles to acquire infectious potential. Furthermore, we clearly show that infection alters the morphology of EVs. However, the biological implications of these morphological changes remain to be fully understood. To address this, we have initiated new lines of investigation analyzing the proteomic and lipidomic content of EVs using mass spectrometry. These approaches may help us identify which cellular pathways are regulated by the biomolecular content of EVs derived from infected cells, and in turn, propose mechanisms by which the virus may manipulate these pathways to facilitate infection.
3.- Which are the advantages of your approach compared to the state-of-the-art proteomic analysis?
The use of two-dimensional gel electrophoresis (2D-PAGE) in this study enabled a comparative analysis of the protein expression profile in HUVEC cells infected with Zika virus. Although this classical approach is less sensitive than state-of-the-art proteomic platforms such as mass spectrometry coupled with LC-MS/MS fractionation techniques, it offers important advantages, including the direct visualization of protein expression levels under the evaluated experimental conditions, as well as effective separation based on isoelectric point and molecular weight. This provides a practical means of characterizing the identity of selected candidate proteins. Furthermore, 2D-PAGE remains one of the most economically accessible methods and offers versatile, global insight into protein expression dynamics. In our infected samples, differential expression was observed in proteins associated with oxidative stress, autophagy, and exosome biogenesis. Unlike high-resolution methods that can generate large and complex datasets requiring advanced bioinformatics tools for biological interpretation, 2D-PAGE provides a more straightforward and accessible view of expression patterns. Nevertheless, the findings obtained through this method serve as a valuable starting point for more in-depth follow-up studies using high-coverage proteomic techniques to better identify and quantify key proteins involved in ZIKV pathogenesis.
4.- Line 49 – “(…) condition. Interestingly, (…)”
We appreciate the observation of the reviewer and we have made the changes related to this comment.
5.- Line 55 – The reviewer suggests adding Western Blot as a keyword since the proteomics is performed using this technique.
We appreciate the observation of the reviewer and we have made the changes related to this comment.
6.- Line 75 – “(…) extracellular vesicles (EVs), currently 75 markers are used to discriminate (…)” to “(…) vesicles (EVs). Currently (…)”
We appreciate the observation of the reviewer and we have made the changes related to this comment.
7.- Line 75 – 84 and line 85 – 93 – The references only in the end does not seems correct. Use them along the paragraph
We appreciate the observation of the reviewer and we have made the changes related to this comment.
8.- Line 75 – Is the work on exosomes or EVs in general?
It is focused on EVs, but there is evidence with exosomal markers that certain isolates we make from EVs have exosomes
9.-Line 131 “(…) (INPer), the project (…)” to “(…) (INPer). The project (…)
We appreciate the observation of the reviewer and we have made the changes related to this comment.
10.- Line 145 – Add where the Vero Cells came from.
We appreciate the observation of the reviewer and we have made the changes related to this comment.
11.-Line 164 – Objective and lasers used
We appreciate the observation of the reviewer and we have made the changes related to this comment.
12.-Line 165 – Is there any special FBS used to isolate exosomes, such as the use of a FBS exosomes-free?
it has been reported that FBS contains exosomes; therefore, to eliminate these and avoid interference with experimental results, it is recommended to ultracentrifuge and filter the FBS to deplete extracellular vesicles (EVs). This procedure is widely recommended and routinely performed in most studies investigating EVs.
13.- Line 174 – Any quantification on the amount obtained?
Quantification is carried out in the NTA after obtaining the samples by ultracentrifugation.
14.- Line 176 – What is a “cell package”?
We appreciate the observation of the reviewer and we have made the changes related to this comment.
15.- Line 178 – hour is h not H
We appreciate the observation of the reviewer and we have made the changes related to this comment.
16.-Line 176 – Add the origin of the antibodies and the reactivity (human, goat, mouse, rat,…)
We appreciate the observation of the reviewer and we have made the changes related to this comment.
17.- Line 185 – How many events were acquired?
We appreciate the observation of the reviewer and we have made the changes related to this comment.
18.-Line 256 – HUVEC
We appreciate the observation of the reviewer and we have made the changes related to this comment.
19.- Line 297 – Infected mock cells were used as a control? Or cells not infected?
We appreciate the observation of the reviewer and we have made the changes related to this comment. We used in assays virus inactivated with UV.
20.-Line 574 – Is this the novelty of this work? A proteomic analysis was never performed in HUVEC cells infected with ZIKV?
The novelty of this study lies in the analysis of exosomes derived from ZIKV-infected HUVEC cells, including changes in the expression of CD9 and Alix proteins. Most importantly, it provides the first evidence that infection by this flavivirus influences the morphological architecture of EVs, offering strong support that such changes are not exclusive to non-infectious diseases.
21.- Line 608—The reviewer suggests adding a paragraph outlining the study's limitations. Nowadays, proteomics relies on very sophisticated methodologies involving LCMS, which are more precise.
Thank you for your comment. We have addressed it and made the corresponding changes, which can be seen highlighted in yellow in the revised version of the manuscript.
Finally, we again thank you for your suggestions and insights, which have enriched the manuscript and produced a more balanced and better account of the review. We hope that the revised manuscript is now suitable for publication in the prestigious journal that you represent.
I look forward to your reply.
Sincerely,
Dr. León-Juárez Moisés
Departamento de Inmunobioquímica
Instituto Nacional de Perinatología ‘Isidro Espinosa de los Reyes’
Montes Urales #800, Col. Lomas de Virreyes
CP 11000, Ciudad de México, México.
01 55 5520 9900 ext-438
E-mail: moisesleoninper@gmail.com

Round 2
Reviewer 1 Report
Comments and Suggestions for Authors
none
Reviewer 3 Report
Comments and Suggestions for Authors
The authors replied successfully to all my comments